# What Should We Pay More Attention to Marfan Syndrome Expecting Ectopia Lentis: Incidence and Risk Factors of Retinal Manifestations

**DOI:** 10.3390/jpm13030398

**Published:** 2023-02-24

**Authors:** Yan Liu, Tianhui Chen, Yongxiang Jiang

**Affiliations:** 1Eye Institute and Department of Ophthalmology, Eye & ENT Hospital, Fudan University, Shanghai 200031, China; 2Key Laboratory of Myopia of State Health Ministry (Fudan University), Key Laboratory of Myopia, Chinese Academy of Medical Sciences, Shanghai 200031, China; 3Shanghai Key Laboratory of Visual Impairment and Restoration, Shanghai 200031, China

**Keywords:** Marfan syndrome, posterior staphyloma, retinal detachment, maculopathy, ATN

## Abstract

(1) Background: This paper investigates the incidence and risk factors of retinal manifestations in patients with Marfan syndrome (MFS) in a Chinese cohort. (2) Methods: This is a population-based cross-sectional study. In total, 344 eyes (172 MFS participants) were enrolled, each of whom underwent a detailed ocular examination. B-scan ultrasonography, ultra-wide-angle fundus images and optical coherence tomography images were conducted to assess posterior staphyloma, types of retinal damages and maculopathy. (3) Results: MFS patients have a high proportion (32.5%) of maculopathy, among which atrophy is the most common type (27.6%). Compared with participants without maculopathy, participants with maculopathy had a longer axial length (AL), higher incidence of posterior staphyloma, macular split and retinal detachment (RD) (*p* < 0.001, *p* < 0.001, *p* < 0.001 and *p* = 0.001). Moreover, the stage of RD has a significant correlation with longer AL and shallower anterior chamber depth (ACD) (*p* = 0.001 and *p* = 0.034, respectively). (4) Conclusions: A higher incidence and earlier onset of fundus lesions were found in MFS patients. Yearly systematic examination is recommended for MFS children with fundus manifestation until the cardiovascular and skeletal development is complete.

## 1. Introduction

Marfan syndrome (MFS), typically caused by mutations in the FBN1 gene encoding fibrillin-1, which plays important roles in systemic connective tissues and affects the integrity and function of extracellular matrix protein, is a multisystem inherited disorder of autosomal dominant inheritance. It has an estimated morbidity rate ranging from 4.60 to 6.81 per 100,000 people [1,2,3]. MFS has a high degree of clinical variability that affects many parts of the body such as the cardiovascular, musculoskeletal, and ocular systems.

The ocular manifestations of MFS include anterior segment changes and posterior segment lesions, which may be due to fibrillin dysplasia. In addition to ectopia lentis (EL), the anterior segment ocular abnormalities in MFS patients comprise flat cornea, deep anterior chamber, and increased susceptibility to cataracts and glaucoma. The posterior segment ocular abnormalities include vitreous liquefaction, lattice degenerations, posterior staphyloma, retinal degeneration, retinal tears and retinal detachment (RD), all of which may worsen the quality of vision [4,5,6]. Previous studies have demonstrated that the incidence of posterior segment lesions is higher than in the general population, occurring up to 18% of eyes of MFS patients, and the incidence is even higher (70%) of MFS patients with subluxated lens [7]. RD is the most serious ocular manifestation of MFS, with an incidence ranging from 5% to 11%; it increases to 8% to 38% in the presence of lens dislocation [8]. What is more, according to a recent nationwide epidemiological study, the incidence of RD in MFS patients has significantly increased compared with the previous studies. Long axial length (AL), early vitreous liquefaction, abnormal peripheral vitreoretinal adhesions, and EL causing vitreous traction seem to be associated risk factors for RD, resulting in a common trend of a giant retinal tear or multiple retinal holes in the peripheral retina [5,9,10,11,12]. The most common site of the retinal breaks was the superior and temporal quadrants [13]. Congenital EL is the most common form of lens dislocation. It is the second leading cause of lens surgery in children after congenital cataracts, and it affects an estimated 6.4/100,000 children. According to our research, EL is one of the most important factors affecting fundus health, and a deep understanding of its mechanisms can help us better prevent it. After studying the susceptibility factors and age of fundus diseases, we can conduct more targeted examinations of the population and reduce the missed diagnosis rate.

However, retinal breaks or early RD in these patients can be missed on routine examinations, owing to lens abnormalities and poor visualization of the fundus. The formation of posterior staphyloma can lead to other damage such as atrophy, tractional, or neovascular maculopathy [14,15]. Macular degeneration is significantly more common in MFS patients. Best-corrected visual acuity (BCVA) has been reported to be worse in eyes with posterior staphyloma [15,16,17]. Most of the changes may threaten vision as they often lead to serious damage to the retinal photoreceptors.

To our knowledge, RD is one of the most common sight-threatening conditions that can lead to blindness without surgical intervention. Moreover, there is no management available yet for the above complications, and since it is rare for eyes to regain normal vision after treatment, preventive measures to avoid the development are warranted. Prompt prophylactic approaches to retinal holes and tears are crucial in preventing RD, in which interventions are often effective. The timely identification and intervention for fundus lesions can greatly improve a patient’s visual quality.

Even though substantial effort has been made to explore the anterior segment manifestations of MFS, there is not yet a relatively not comprehensive and profound understanding of the posterior segment lesions. Most of the previous studies were based on empirical conclusions from clinical observations, there was no specific analysis of the type of complication and incidence of each subtype, and there was a lack of children in the study participants, so it is important to expand the sample size. Moreover, elucidating the characteristics of ocular parameters and risk factors associated with posterior lesions in MFS patients is extremely essential to preserve patients’ visual function. In this study, we intended to analyze the characteristics and investigate the possible correlations between ocular parameters (AL, BCVA, mean Keratometry (Km), anterior chamber depth (ACD)) and posterior segment lesions in a Chinese cohort of MFS. A range of ocular comorbidities (posterior staphyloma, maculopathy, macular split and RD) were compared for different subgroups. The results revealed the posterior segment characteristics of MFS, screened for related parameters, and provided information for further studies.

## 2. Materials and Methods

The study was performed according to the Declaration of Helsinki and approved by the Human Research Ethics Committee of the Eye & ENT Hospital of Fudan University. This is a population-based cross-sectional study. Written informed consent was obtained from all patients or the patients’ guardians for those under 18 years of age.

Inclusion/Exclusion Criteria: Patients diagnosed with MFS based on the Ghent-2 criteria from 29 January 2013 to 19 March 2022 were eligible. Both eyes of each participant were included. Patients with a history of trauma and incomplete eye image and examination data were excluded from the study. The study group comprised 172 MFS patients (344 eyes).

Classification Criteria: The groups were divided as follows: those with spherical equivalent refraction (SER) ≥ −6 diopters (D) or an AL ≥ 26 mm were included in the Pathological myopic group, and the remaining participants were included in the Non-pathological myopic group. The classification for grouping by different maculopathy types was based on the new classification and grading system (ATN) for myopic maculopathy, which considers atrophy (A), tractional (T), and neovascular (N) components [18,19]. The eyes with a grade of A1 or higher were classified as atrophic maculopathy. The eyes with a grade of T1 or higher were classified as tractional maculopathy. The eyes with a grade of N1 or higher were classified as neovascular maculopathy. The staging criteria for retinal detachment are as follows: stage 0 indicates the absence of retinopathy, stage 1 indicates retinal atrophy, stage 2 indicates atrophy of the retina with macular split, stage 3 indicates RD without macula involvement, stage 4 indicates RD involving the macula, and stage 5 indicates full-thickness RD [20]. These classification results were mainly obtained by clinicians based on a review of fundus photographs and supplemented by clinical examination.

Ophthalmic examination: All participants were examined by a board-certified optometrist and ophthalmologist, and their medical history was carefully evaluated. The BCVA was converted into the logarithm of minimal angle of resolution (logMAR). Ectopia lentis (EL) was diagnosed as the dislocation of the lens from its anatomical place. The range of EL dislocation was assessed by the curvature degree (angle α) of the ring (pupil)-ring (lens) cross under complete pupillary dilation (Figure 1) [21]. The AL and ACD were determined using partial coherence interferometry (IOL Master 500 and 700; Carl Zeiss Meditec, Jena, Germany), and the Km (D) was taken as the corneal curvature. The posterior staphyloma was detected by B-scan ultrasonography. Retinal photographs centered on the macular and optic disc were obtained for both eyes using the ultra-wide angle fundus images (SPECTRALIS; Heidelberg Engineering, Heidelberg, Germany).

Statistical analyses: SPSS 23.0 (IBM Corp., Armonk, NY, USA) was used for all the statistical analyses. The Shapiro–Wilk test was used to confirm whether the variables followed normal distributions. The Z-scores of AL was calculated via the following formula: Z-score = (measured parameter—normative parameter)/normative standard deviation [22]. Demographic and ocular characteristics were compared across groups using the Wilcoxon rank-sun test (Mann–Whitney test), Kruskal–Wallis test, Student’ s *t*-test and Chi-square test. Correlations between the presence or absence of pathological myopia, AL, ACD, Km and the stage of RD were assessed by calculating Pearson’s correlation coefficient (r). The proportion of participants in different maculopathy grades is presented by the sunburst chart. Multivariate linear regression model was constructed with atrophy, traction, or neovascularization as the dependent variable to assess the association with the status. Odds ratios (ORs) and 95% confidence intervals (Cis) were calculated, and all *p* values were 2-sided and considered statistically significant when they were less than 0.05.

## 3. Results

A total of 344 eyes of 172 participants with MFS were included in this study. Table 1 summarized the demographic and clinical characteristics of these participants. The mean age was 13.66 ± 12.60 and more than 70% of patients were younger than 20 years. The eyes had longer AL, deeper ACD, and flatter Km than the normal reference ranges. The average BCVA (logMAR) was 0.81 ± 0.47, which demonstrated worse visual quality than normal population.

In Table 2, participants were divided into two groups based on whether they had pathological myopia. The Pathological myopic group tended to be older (*p* < 0.001). Patients in the Non-pathological myopic group were less likely to have systemic manifestations than those in the Pathological myopic group (*p* = 0.036). There were no significant differences in sex, intraocular pressure (IOP), ACD and the range of lens dislocation between the Pathological myopic group and the Non-pathological myopic group. Regarding the direction of lens dislocation, both groups showed a tendency of being more prone to shift toward the nasal and temporal sides. However, it displayed a significant difference in the proportion of different directions (*p* < 0.001) (Figure 2). Furthermore, the Pathological myopic group tended to have worse visual quality (*p* < 0.001), longer AL (*p* < 0.001), as well as longer Z-AL (*p* < 0.001). Participants in the Pathological myopic group had a higher incidence of posterior staphyloma (43.9% vs. 18.1%, *p* < 0.001) and RD (14.0% vs. 5.2%, *p* = 0.013). There was a significant difference in the proportion of RD stage between the Pathological myopic group and the Non-pathological myopic group (*p* = 0.004). After analyzing the association of ocular parameters such as age, Z-AL, BCVA, Km, ACD and IOP with RD stage, we found that participants with longer Z-AL and shallower ACD were significantly more likely to have more severe RD (*p* = 0.001 and *p* = 0.034) (Table 3). Figure 3 shows four eyes had an AL longer than 28 mm, with multiple retinal breaks, RD and giant retinal tear, respectively.

Maculopathy including atrophy, traction, and neovascularization was then analyzed for the different parameters and ocular comorbidities of the eyes with pathological myopia (Table 4). The prevalence of eyes with atrophy was higher than that of eyes with traction and neovascularization (27.6%, 7.5% and 7.9%, respectively). The prevalence of eyes with maculopathy was 32.5% in 228 eyes with pathological myopia. Male and older participants had a greater likelihood that suffering from more severe fundus lesions. Patients with cardiac disease are most likely to have more server macular degeneration. Compared with participants without maculopathy, participants with atrophy, tractional or neovascular maculopathy had worse BCVA and longer AL. Furthermore, participants with tractional maculopathy had worse vision and longer AL. No significant differences were found in IOP, ACD, Km, the classification of lens dislocation direction and range between these four subgroups. Participants without maculopathy had a much lower incidence of posterior staphyloma, macular split and RD compared with those with maculopathy, especially tractional maculopathy.

Figure 4 shows the distribution of different categories of atrophy, traction and neovascularization. Of the eyes with maculopathy, 11.84%, 38.36%, 50.68%, 2.74%, and 0.00% had grades of A0, A1, A2, A3 and A4, respectively, 77.63%, 7.89%, 0.00%, 9.21%, 3.95%, and 1.32% had grades of T0, T1, T2, T3, T4 and T5, respectively; 76.32%, 21.05% and 2.63% had grades of N0, N2a and N2s, respectively. In eyes with atrophic maculopathy, 77.89%, 7.14%, 21.62%, 0.00% and 0.00% had tractional maculopathy, while 44.44%, 14.29%, 27.03%, 0.00% and 0.00% had neovascular maculopathy. The proportion of traction and neovascularization increased with the increased severity of atrophy, but the severity of traction and neovascular maculopathy was not precisely consistent with the severity of atrophy. Figure 5 shows the associations of age, AL, the presence of posterior staphyloma, RD and dislocation range with maculopathy in multivariate linear models. Eyes with longer AL and greater dislocation range were more likely to have atrophic maculopathy. The presence of RD was significantly associated with atrophic maculopathy (OR, 8.861; 95% CI, 1.299~60.444; *p* = 0.026). The participants with posterior staphyloma were more likely to have atrophic maculopathy (OR, 0.007, 95% CI, 1.031~9.159, *p* < 0.001). No significant associations were found between age and maculopathy.

## 4. Discussion

MFS is often combined with posterior segment alternations, especially the fundus lesions, resulting in poor visual quality. The identification of factors responsible for fundus lesions is critical. However, few studies have reported the characteristics of MFS’ s posterior segment. In this retrospective analysis of the Chinese cohort with MFS, we mainly analyzed the posterior segment manifestations of MFS and identified factors associated with the occurrence of posterior staphyloma, maculopathy and RD [23,24]. Patients in our cohort had a higher incidence of posterior staphyloma (35.2% vs. 13.2%) than those young highly myopic patients with ages ranging between 3 and 19 years [25,26]. Compared with the general and myopic population, MFS patients show higher incidence and earlier onset of fundus lesions.

Myopia is the most prevalent ocular disorder in MFS patients that develops rapidly in early childhood. Previous studies elucidated that the prevalence of myopia in MFS (34~44%) is higher than in the general population (4.8%) [5,7]. In our cohort, the prevalence of pathological myopia was up to 47%. Meanwhile, the incidence of ocular comorbidities was higher in patients with MFS than in people with general high myopia. In this study, we further observed that the stage of RD in patients with MFS was more serious than in those without maculopathy. Maculopathy was found in 32.5% of MFS patients in the Pathological myopic group. Atrophic maculopathy was the predominant type, which was followed by tractional and neovascular maculopathy (27.6%, 7.5% and 7.9%, respectively). The mechanism of neovascular maculopathy accounts for a higher proportion than tractional maculopathy may be explained by that neovascularization tends to develop at a relatively young age, and our cohort comprised largely participants younger than 20 years old. Significant differences were found in BCVA and AL among the four groups. longer AL and worse BCVA appeared to show in MFS patients with tractional maculopathy as well as atrophic and neovascular maculopathy.

The incidence of RD was 11.0% in our cohort, which is consistent with the conclusion of the previous study (8~25.6%) [9,27], showing patients with MFS were more likely to be complicated with RD, dramatically at a younger age around 20 years old. Several studies showed that the peripheral retinal changes in MFS include myopic degeneration, lattice degeneration, atrophic holes, chorioretinal pigment proliferation, peripheral vitreous traction syndromes, and retinal breaks [5]. As for the risk factors, in addition to lens dislocation and longer AL, there also include early vitreous liquefaction, posterior vitreous detachment without any dehiscence at the vitreoretinal interface, and abnormal peripheral vitreoretinal adhesions [5]. Extensive vitreous liquefaction in central and peripheral areas is a common finding in patients with MFS. Additionally, vitreous attachment along the edges of peripheral retinal abnormalities such as lattice degeneration is a common finding in these patients. Vitreous degeneration and vitreous hemorrhage were also significantly more common in MFS compared with the controls. The increased AL in patients with MFS with a dislocated lens increases the risk of having vitreous degeneration or hemorrhage [7].

The history of intraocular surgery is also a major risk factor for RD in MFS. Previous studies have reported a 25% incidence of RD after surgery for ectopia lentis [9]. Meanwhile, another study showed that 37% of RD patients had a history of cataract extraction with intraocular lens implantation [28]. Patients with MFS have a significantly increased risk of RD because of EL and intraocular surgery. Hence, there is a need for ophthalmologists to pay more attention to the preoperative ocular parameters and postoperative follow-up of fundus changes in MFS patients.

Defects in the FBN1 (Fibrillin 1) gene on chromosome 15 is the cause of MFS. FBN1 is widely distributed in ocular tissues, providing force-bearing structural support and elasticity to the ocular connective tissues. Aside from zonules, FBN1 has also been localized on the lens capsule, sclera and Bruch’s membrane. The genetic defect also can explain the weakness in the scleral structure and tensile strength, leading to the associated retinal complications [9,13]. FBN1 is found abundantly in the healthy sclera. In MFS, the sclera is very thin and collapsible, with loss of elasticity and stretching leading to long AL and posterior segment lesions [13]. The misfolded FBN1 and the disorganized elastin in the scleral stroma could lead to stretching of the sclera through the two physiological functions of fibrillin-1 (structure component or TGF-β signaling mediator) and by different underlying genetic mechanisms (haploinsufficiency or dominant-negative effect) [22].

These changes result from the formation of posterior staphyloma, and it has been suggested that the staphyloma causes mechanical tension on the retina, which then leads to damage. Posterior staphyloma tends to increase in size causing an increase in the AL and thus a myopic shift [29]. Posterior staphyloma is strongly associated with macular changes [30].

Systemic symptoms, including cardiac manifestations and the skeletal system was also found to be related to fundus change. Yearly echocardiography and X-ray on the skeletal system are recommended for MFS children with fundus manifestation until the completed cardiovascular and skeletal development.

There were some limitations in our study. First of all, it is a retrospective methodology, and the lack of a general control group for comparison might have resulted in bias in our findings. A longitudinal study for long-term follow-up is needed for the development of fundus changes.

## 5. Conclusions

In conclusion, this study enabled us to better understand the pathophysiological fundus findings in MFS patients. This study represents the largest such analysis of posterior segment lesions for MFS patients in an Asian population to our knowledge. Overall, a higher incidence and earlier onset of fundus lesions were found in MFS patients. Longer AL and worse BCVA are risk factors for posterior staphyloma and RD. Furthermore, longer AL, older age, steeper Km, shallower ACD and posterior staphyloma formation predict more severe RD, while the incidence of maculopathy is also high in MFS patients. Annual systematic examinations of MFS children with fundus manifestations are warranted, investigating the manifestations of MFS patients will allow timely and accurate awareness of the incidence and risk factors of retinal lesions.

## Figures and Tables

**Figure 1 jpm-13-00398-f001:**
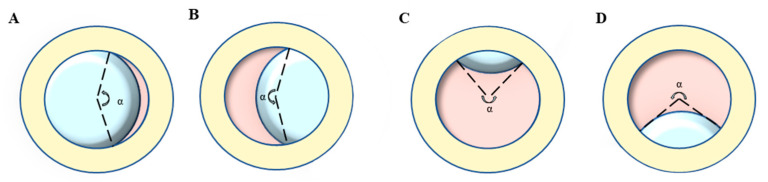
Lens dislocation direction and the representation of curvature degree (angle α): take the left eye of the opposite patient as an example. (**A**) Nasal direction, (**B**) Temporal direction, (**C**) Superior direction, (**D**) inferior direction, (**E**) Anterior direction, (**F**) Posterior direction.

**Figure 2 jpm-13-00398-f002:**
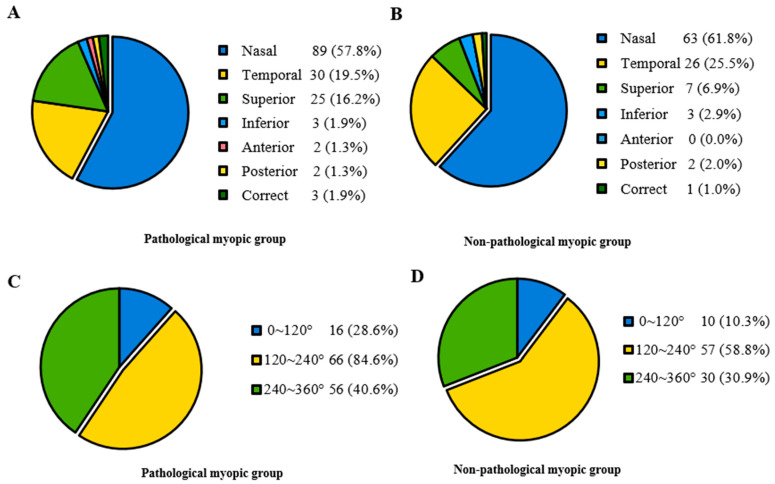
The proportion of lens dislocation direction and range in the Pathological myopic and Non-pathological myopic groups. (**A**) Lens dislocation direction in the Pathological myopic group, revealing most participants displayed nasal, temporal and superior lens displacements. (**B**) Lens dislocation direction in the Non-pathological group, revealing most participants displayed nasal, temporal and superior lens displacements. (**C**) Lens dislocation range in the Pathological myopic group, revealing most participants displayed moderate to severe range. (**D**) Lens dislocation range in the Non-pathological myopic group, revealing most participants displayed moderate to severe range.

**Figure 3 jpm-13-00398-f003:**
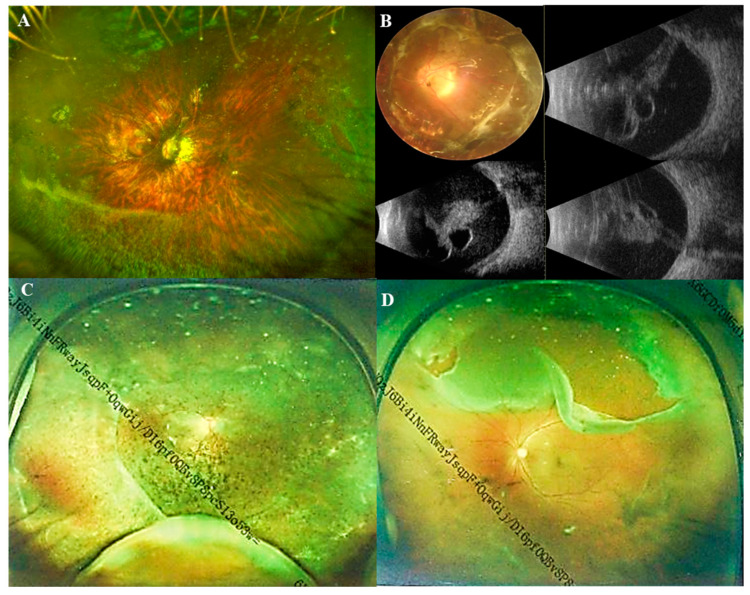
Ultra-wide-angle fundus images and B-scan ultrasonography images of Marfan patients. (**A**) Ultra-wide-angle fundus image of a pathological myopic eye with nasal retinal breaks and a horseshoe tear in superotemporal; (**B**) A fundus image and B-scan ultrasonography images of a pathological myopic eye with a total retinal detachment; (**C**) Ultra-wide-angle fundus image of a pathological myopic eye with retinal detachment in superotemporal, inferotemporal, and inferonasal quadrants; (**D**) Ultra-wide-angle fundus image of a pathological myopic eye with a giant retinal tear above.

**Figure 4 jpm-13-00398-f004:**
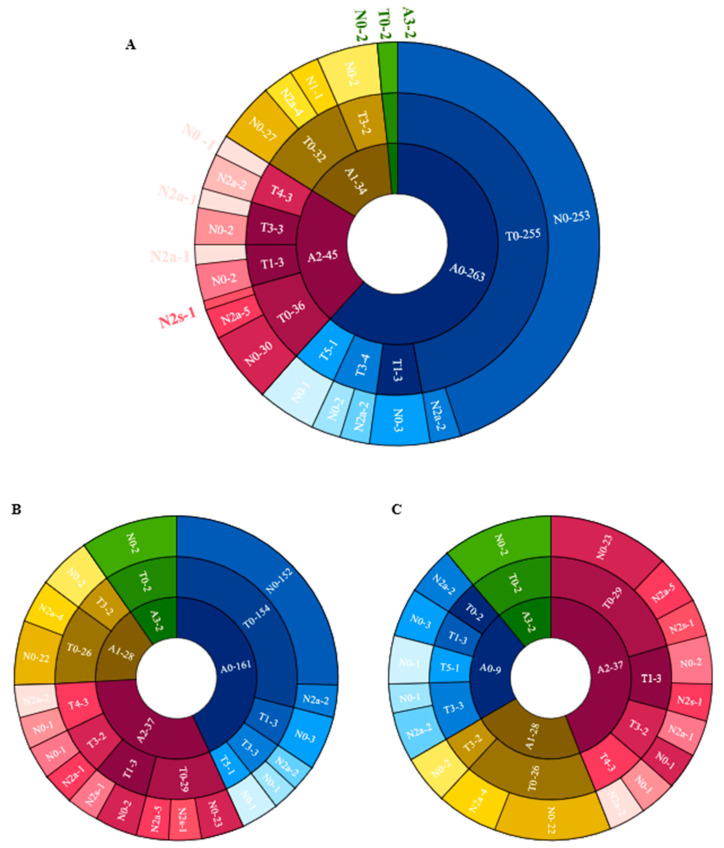
The distribution of different categories of atrophy, traction and neovascularization. (**A**) The distribution of different categories of atrophy, traction and neovascularization in the total cohort. (**B**) The distribution of different categories of atrophy, traction and neovascularization in the Pathological myopic group. (**C**) The distribution of different categories of atrophy, traction and neovascularization in participants with maculopathy.

**Figure 5 jpm-13-00398-f005:**
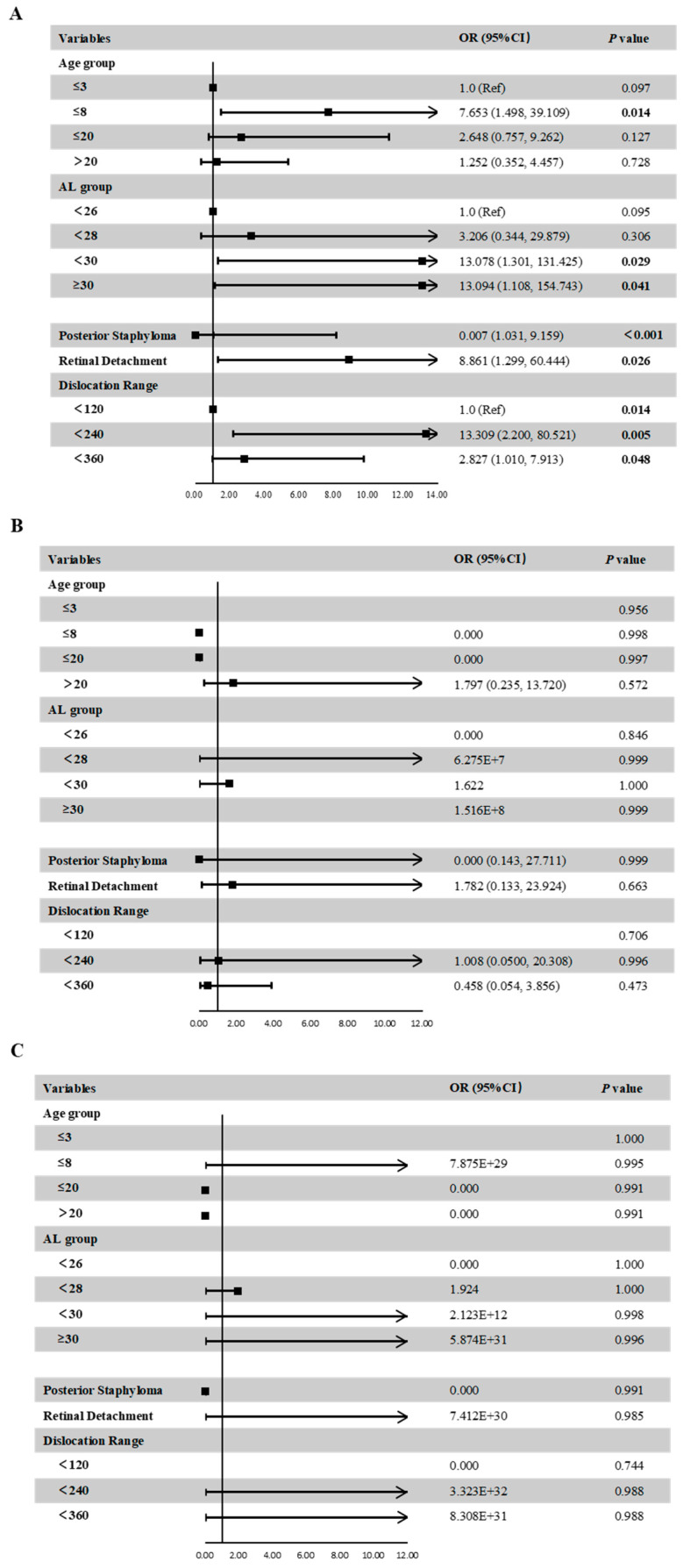
Associations between ocular parameters and the presence of atrophic maculopathy (**A**), tractional maculopathy (**B**), or neovascular maculopathy (**C**) in the multivariate adjusted models.

**Table 1 jpm-13-00398-t001:** The demographic status of the 172 participants with MFS.

Characteristics	Mean ± Standard Deviation (Range) or Number (%)
Eyes	344 (100%)
Age (year)	13.66 ± 12.60
≤3	52 (15.1%)
≤8	132 (38.4%)
≤20	62 (18.0)
>20	98 (28.5%)
Systemic Manifestation	
Cardiovascular	8 (2.3%)
Musculoskeletal	6 (1.7%)
Central Nervous	2 (0.6%)
No	328 (95.3%)
Dislocation Direction	
Nasal	152 (59.4%)
Temporal	56 (21.9%)
Superior	32 (12.5%)
Inferior	6 (2.3%)
Anterior	2 (0.8%)
Posterior	4 (1.6%)
Correct	4 (1.6%)
Dislocation Range	
0~120°	26 (11.1%)
120~240°	123 (52.3%)
240~360°	86 (36.6%)
BCVA (logMAR)	0.81 ± 0.47
AL (mm)	25.28 ± 3.23
Z-AL	2.92 ± 3.45
IOP (mmHg)	14.86 ± 4.91
ACD (mm)	3.13 ± 0.60
Km (D)	39.81 ± 1.74
Posterior Staphyloma	121 (35.2%)
Macular Split	5 (1.5%)
Retinal Detachment	38 (11.0%)

AL = axial length; ACD = anterior chamber depth; BCVA = best-corrected visual acuity; IOP = intraocular pressure; logMAR = logarithm of minimal angle of resolution; Km = mean Keratometry.

**Table 2 jpm-13-00398-t002:** Baseline characteristics for MFS in the Pathological myopic group and the Non-pathological myopic group.

	Pathological Myopic Group	Non-Pathological Myopic Group	*p* Value
Eyes	228 (66.3%)	116 (33.7%)	
Male, n (%)	143 (62.7%)	81 (69.8%)	0.191
Age (year)	16.51 ± 13.33	8.22 ± 8.83	<0.001 *
Age group			<0.001 *
≤3	25 (11.0%)	27 (23.3%)	
≤8	66 (28.9%)	66 (56.9%)	
≤20	51 (22.4%)	11 (9.5%)	
>20	86 (37.7%)	12 (10.3%)	
Systemic Manifestation			0.036 *
Cardiovascular	8 (3.5%)	0 (0.0%)	
Musculoskeletal	6 (2.6%)	0 (0.0%)	
Central Nervous	2 (0.9%)	0 (0.0%)	
No	212 (93.0%)	116 (100.0%)	
Len Dislocation Direction			<0.001 *
Len Dislocation Range			0.242
BCVA (logMAR)	0.95 ± 0.42	0.41 ± 0.36	<0.001 *
AL (mm)	26.07 ± 3.39	23.27 ± 1.48	<0.001 *
Z-AL	3.91 ± 4.14	1.54 ± 1.18	<0.001 *
AL group			<0.001 *
<26	72 (31.6%)	116 (100.0%)	
<28	45 (19.8%)	0 (0.0%)	
<30	31 (13.6%)	0 (0.0%)	
≥30	25 (11.0%)	0 (0.0%)	
IOP (mmHg)	15.05 ± 3.10	14.33 ± 2.90	0.450
ACD (mm)	3.10 ± 0.52	3.13 ± 0.52	0.546
Posterior Staphyloma	100 (43.9%)	21 (18.1%)	<0.001 *
Macular Split	5 (2.2%)	0 (0.0%)	0.108
Retinal Detachment	32 (14.0%)	6 (5.2%)	0.013 *

AL = axial length; ACD = anterior chamber depth; BCVA = best-corrected visual acuity; IOP = intraocular pressure; logMAR = logarithm of minimal angle of resolution; * *p* < 0.05.

**Table 3 jpm-13-00398-t003:** The proportion of retinal detachment stage in the Pathological myopic group and the Non-pathological myopic group.

	Pathological Myopic Group	Non-Pathological Myopic Group	Total
Retinal Detachment Stage	0	140 (58.6%)	96 (76.9%)
	1	51 (21.6%)	13 (14.3%)
	2	5 (1.9%)	1 (1.6%)
	3	20 (11.1%)	5 (4.9%)
	4	6 (3.1%)	1 (2.2%)
	5	6 (3.7%)	0 (0.0%)
Total		228 (100.0%)	116 (100.0%)

**Table 4 jpm-13-00398-t004:** Characteristics and Comparison among Pathological Myopic Eyes with Three Different Maculopathy Types.

	None	Atrophy	Traction	Neovascularization	*p* Value
Eyes	154 (67.5%)	63 (27.6%)	17 (7.5%)	18 (7.9%)	
Male, n (%)	86 (55.8%)	46 (73.0%)	15 (88.2%)	16 (88.9%)	0.001 *
Age (year)	17.67 ± 13.51	11.92 ± 13.21	24.94 ± 13.46	23.78 ± 8.85	0.002 *
Age group					0.001 *
≤3	16 (10.4%)	9 (14.3%)	0 (0.0%)	0 (0.0%)	
≤8	51 (33.1%)	15 (23.8%)	0 (0.0%)	0 (0.0%)	
≤20	30 (19.5%)	18 (28.6%)	6 (35.3%)	8 (44.4%)	
>20	57 (37.0%)	21 (33.3%)	11 (64.7%)	10 (55.6%)	
Systemic Manifestation					<0.001 *
Cardiovascular	4 (2.6%)	4 (6.3%)	1 (5.9%)	2 (11.1%)	
Musculoskeletal	0 (0.0%)	2 (3.2%)	4 (23.5%)	2 (11.1%)	
Central Nervous	0 (0.0%)	2 (3.2%)	0 (0.0%)	0 (0.0%)	
No	150 (97.4%)	55 (87.3%)	12 (70.6%)	14 (77.8%)	
BCVA (logMAR)	1.01 ± 0.24	1.06 ± 0.31	1.11 ± 0.39	0.95 ± 0.49	0.499
AL (mm)	23.04 ± 1.36	24.10 ± 1.67	/	22.00 ± 1.73	<0.001 *
Z-AL	1.97 ± 2.13	4.33 ± 2.78	6.58 ± 3.33	8.09 ± 3.73	<0.001 *
IOP (mmHg)	15.28 ± 5.61	14.60 ± 3.59	13.82 ± 2.46	15.69 ± 8.12	0.908
ACD (mm)	3.03 ± 0.53	3.21 ± 0.49	3.27 ± 0.72	3.17 ± 0.17	0.118
Km (D)	39.96 ± 1.97	40.081.85	40.41 ± 2.50	39.65 ± 0.99	0.908
Posterior Staphyloma	47 (30.5%)	44 (69.8%)	14 (82.4%)	18 (100%)	<0.001 *
Macular Split	0 (0.0%)	3 (4.8%)	5 (29.4%)	2 (11.1%)	<0.001 *
Retinal Detachment	14 (9.1%)	13 (20.6%)	7 (41.2%)	6 (33.3%)	0.001 *

AL = axial length; ACD = anterior chamber depth; BCVA = best-corrected visual acuity; IOP = intraocular pressure; logMAR = logarithm of minimal angle of resolution; Km = mean Keratometry. * *p* < 0.05.

## Data Availability

The data presented in this study are available on request from the corresponding author.

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
