# Peer review of "What Should We Pay More Attention to Marfan Syndrome Expecting Ectopia Lentis: Incidence and Risk Factors of Retinal Manifestations"

_jpm, 2023, doi:10.3390/jpm13030398_

Round 1
Reviewer 1 Report
The authors attempted to investigate the incidence and risk factors of retinal manifestations in patients with Marfan syndrome in a Chinese cohort, I appreciate their efforts. However, several questions must be answered and discussed in the manuscript.
At first, it is impressive that the authors recruited so many young patients with Marfan syndrome (MFS), Even though they had commented that subjects were diagnosed with MFS based on the Ghent-2 criteria, they needed to show the detailed information about FBN1 gene analysis and what kind of cardiovascular manifestations that the subjects had. According to the Ghent-2 criteria, aortic root aneurysm and ectopia lentis (dislocated lenses) are cardinal features of MFS. It was quite strange that the subjects with cardiovascular manifestation were 8 (2.3%) patients in their cohort.
Additionally, the authors needed to declare how they had diagnosed four patients without dislocated lenses in table 1 as MFS,
There are several disease entities showing dislocated lenses at young age such as lens coloboma, homocystinuria, Weill-Marchesani syndrome, and Ehlers-Danlos syndrome. I wondered if the authors completely excluded those patients in their cohort.
Reviewer 2 Report
It would be interesting to see how the general population in the same place measures as a reference.
But then their study design is a survey and that cannot be a part of this. It would be interesting to see the comparison in some future study.
Reviewer 3 Report
This is a very interesting study and a well-written manuscript. The authors present the incidence and risk factors of retinal manifestations in a large number of patients with Marfan syndrome. The results are well-presented and I find the tables as well as the figures very useful. A spell check is required.
